# Longitudinal and Multimodal Radiomics Models for Head and Neck Cancer Outcome Prediction

**DOI:** 10.3390/cancers15030673

**Published:** 2023-01-21

**Authors:** Sebastian Starke, Alexander Zwanenburg, Karoline Leger, Klaus Zöphel, Jörg Kotzerke, Mechthild Krause, Michael Baumann, Esther G. C. Troost, Steffen Löck

**Affiliations:** 1Helmholtz-Zentrum Dresden–Rossendorf, Department of Information Services and Computing, 01328 Dresden, Germany; 2OncoRay—National Center for Radiation Research in Oncology, Faculty of Medicine and University Hospital Carl Gustav Carus, Technische Universität Dresden, Helmholtz-Zentrum Dresden–Rossendorf, 01309 Dresden, Germany; 3German Cancer Research Center (DKFZ), Heidelberg and German Cancer Consortium (DKTK) Partner Site, 01307 Dresden, Germany; 4National Center for Tumor Diseases (NCT), Partner Site Dresden of the German Cancer Research Center (DKFZ), Faculty of Medicine and University Hospital Carl Gustav Carus, Technische Universität Dresden, 01307 Dresden, Germany; 5Department of Radiotherapy and Radiation Oncology, Faculty of Medicine and University Hospital Carl Gustav Carus, Technische Universität Dresden, 01309 Dresden, Germany; 6Department of Nuclear Medicine, Faculty of Medicine and University Hospital Carl Gustav Carus, Technische Universität Dresden, 01307 Dresden, Germany; 7Helmholtz-Zentrum Dresden-Rossendorf, PET Center, Institute of Radiopharmaceutical Cancer Research, 01328 Dresden, Germany; 8Klinik für Nuklearmedizin, Klinikum Chemnitz gGmbH, 09116 Chemnitz, Germany; 9Helmholtz-Zentrum Dresden–Rossendorf, Institute of Radiooncology—OncoRay, 01328 Dresden, Germany; 10German Cancer Research Center (DKFZ), Division Radiooncology/Radiobiology, 69120 Heidelberg, Germany; 11German Cancer Consortium, Core Center, 69120 Heidelberg, Germany

**Keywords:** radiomics, head and neck cancer, loco-regional control, survival analysis, computed tomography, positron emission tomography, Cox proportional hazards, longitudinal imaging

## Abstract

**Simple Summary:**

Machine learning based radiomics models for prediction of loco-regional recurrence today mostly rely on features extracted from pre-treatment imaging data. In this work, we investigate the predictive ability of such models when imaging data obtained during the course of treatment are used. This is achieved by extracting features from pre-treatment CT and FDG-PET images as well as from images obtained two (only CT) and three weeks after start of radiochemotherapy. Models comprised of combined features from both modalities and multiple timepoints are evaluated. We confirm that predictive model performance is improved when features from in-treatment imaging are used, finding that CT-based features allow for more accurate risk prediction, while FDG-PET based models are able to stratify patients into low- and high-risk groups more reliably.

**Abstract:**

Radiomics analysis provides a promising avenue towards the enabling of personalized radiotherapy. Most frequently, prognostic radiomics models are based on features extracted from medical images that are acquired before treatment. Here, we investigate whether combining data from multiple timepoints during treatment and from multiple imaging modalities can improve the predictive ability of radiomics models. We extracted radiomics features from computed tomography (CT) images acquired before treatment as well as two and three weeks after the start of radiochemotherapy for 55 patients with locally advanced head and neck squamous cell carcinoma (HNSCC). Additionally, we obtained features from FDG-PET images taken before treatment and three weeks after the start of therapy. Cox proportional hazards models were then built based on features of the different image modalities, treatment timepoints, and combinations thereof using two different feature selection methods in a five-fold cross-validation approach. Based on the cross-validation results, feature signatures were derived and their performance was independently validated. Discrimination regarding loco-regional control was assessed by the concordance index (C-index) and log-rank tests were performed to assess risk stratification. The best prognostic performance was obtained for timepoints during treatment for all modalities. Overall, CT was the best discriminating modality with an independent validation C-index of 0.78 for week two and weeks two and three combined. However, none of these models achieved statistically significant patient stratification. Models based on FDG-PET features from week three provided both satisfactory discrimination (C-index = 0.61 and 0.64) and statistically significant stratification (p=0.044 and p<0.001), but produced highly imbalanced risk groups. After independent validation on larger datasets, the value of (multimodal) radiomics models combining several imaging timepoints should be prospectively assessed for personalized treatment strategies.

## 1. Introduction

With a 5-year survival rate of only about 50% for locally advanced head and neck squamous cell carcinoma (HNSCC) treated with primary radiochemotherapy, methods to improve outcome for this tumor entity are needed [1]. One promising direction towards achieving this goal is treatment individualization, i.e., adapting the treatment to take tumor characteristics and other patient-specific biomarkers into account [2]. Machine learning based radiomics employs medical imaging data as a valuable and non-invasive tool for creating individualized risk predictions for cancer patients [3,4]. It has been applied for several tumor entities and treatment endpoints, such as overall survival (OS) [5,6,7,8], loco-regional tumor control (LRC) [6,9,10], progression-free survival (PFS) [11,12,13], and distant metastasis (DM)-free survival [9,14,15]. In a radiomics approach, routinely obtained medical images such as computed tomography (CT) [6,8,9,16], magnetic resonance imaging (MRI) [17,18,19], or positron emission tomography (PET) scans [10,20,21,22] can be used to compute quantitative features that describe, e.g., a tumor’s size, shape, or texture. These features are then used in the creation of statistical models to predict the endpoint of interest. Recently, neural network-based survival models have been introduced as possible alternatives that have the additional benefit of learning important tumor features automatically; however, these in turn require large amounts of annotated data [23,24,25,26,27,28].

To the best of our knowledge, no model has yet reached a performance level sufficient for clinical acceptability and applicability in HNSCC [29,30]. Recent work on large HNSCC cohorts has shown limited prognostic power of radiomics features computed from pre-treatment CT imaging [8]. These findings may be due to the heterogeneity in the cohorts, or they could indicate that pre-treatment imaging, while highly important for initial cancer diagnosis and treatment planning, may contain limited information for predicting outcomes such as OS and LRC.

Radiotherapy for treating HNSCC is commonly applied to a tumor in daily sessions for several consecutive weeks. Additional imaging can be obtained during that time to assess the need for adapting the treatment plan. Initial studies have shown that longitudinal imaging data can be used to improve prediction with respect to treatment endpoints [15,21,31,32,33,34,35]. These studies quantify the temporal evolution of feature values by computing the difference (delta) [21] or rate of change [32,34] between pre-treatment and in-treatment imaging. For a cohort of HNSCC patients, Sellami et al. [34] used in-treatment cone–beam CT (CBCT) imaging to extract reproducible and informative features by considering only features that had prognostic power at all evaluated timepoints. A final model was then built for a single timepoint only using delta features. The work carried out by Wu et al. [15] focused on prediction of DM for HNSCC patients using pre-treatment and mid-treatment CT features as well as delta features. They found that inclusion of features from both timepoints improved performance compared to a model comprised solely of pre-treatment features.

Here, we combine features from multiple timepoints and two modalities (CT and 18F-FDG-PET) in order to assess any increase in predictive potential compared to pre-treatment imaging for LRC in patients with locally advanced HNSCC who received primary radiochemotherapy. This extends the work by Leger et al. [31], who built radiomics models solely on CT imaging and a single in-treatment timepoint. In addition, we compare our approach to models that exclusively used delta features and to baseline models that used tumor volume as the only predictor.

## 2. Materials and Methods

### 2.1. Patient Cohort

We considered two cohorts of patients with locally advanced HNSCC, each of which received primary radiochemotherapy, originating from different prospective observational imaging trials. Patients in the first cohort were originally enrolled in a study with a focus on longitudinal FMISO-PET imaging [20,22]. Due to the larger number of patients, we used this as the exploratory cohort on which we built our radiomics models. Patients in the second cohort were part of a trial that investigated longitudinal FDG-PET imaging. We used patients from this cohort to independently validate the model’s performance.

The imaging data from both cohorts have been previously described and analyzed by Leger et al. [31]. Ethical approval for the multicentre retrospective analyses of clinical and imaging data was obtained from the Ethics Committee at the Technische Universität Dresden, Germany (EK177042017). All analyses were carried out in accordance with the relevant guidelines and regulations. Informed consent was obtained from all patients.

For our analysis, we used a subset of patients for which all of the following imaging data were available: CT scans taken before the start of treatment (week 0, W0), within the second week of treatment (week 2, W2), and shortly after the third week of treatment (week 3, W3), as well as FDG-PET scans before treatment start and shortly after the third week of treatment (Table 1). Acquisition parameters for CT and FDG-PET images are provided in Appendix A, respectively. A Siemens Biograph 16 PET/CT scanner was used as the imaging device, and CT imaging was performed without any contrast agent. We included 37 patients for exploration and model building and 18 patients for independent validation. Details of the patient cohorts can be found in Table 2. The endpoint of this analysis was LRC, defined as the time between the start of radiochemotherapy and local or regional tumor recurrence with previous objective loco-regional tumor response. To ensure that the recurrence occurred within the irradiated volume, the radiotherapy treatment plan and radiological images of the recurrence (CT or PET/CT) were reviewed for each loco-regional failure.

### 2.2. Image Processing and Feature Extraction

The delineation of the primary gross tumor volume (GTV) was performed using the RayStation treatment planning system (version 8B SP2; RaySearch Laboratories AB, Stockholm, Sweden) on the initial planning CT (W0). The diagnostic FDG-PET images were manually registered to the CT image using rigid registration in the treatment planning system. To provide consistent delineation for the radiomic analyses, all clinical ROIs were inspected and re-contoured if required by a radio-oncologist in training (K.L.) and validated by an experienced radio-oncologist (E.G.C.T.). For W2 and W3, contours from W0 were transferred and adapted to the GTV based on the corresponding CT imaging data. FDG-PET intensities were converted to body weight-corrected standard uptake values (SUV). Patient images were then interpolated to isotropic voxel spacing (1 × 1 × 1 ${\mathrm{mm}}^{3}$ and 3 × 3 × 3 ${\mathrm{mm}}^{3}$ for CT and FDG-PET, respectively) using cubic splines. Segmentation masks were linearly interpolated and binarized using a cutoff of 0.5, and 182 radiomics features capturing statistical, morphological, intensity based and texture based characteristics were computed from each modality at every timepoint using the ‘medical image radiomics processor’ package (MIRP) [36,37]. The bin width was set to 12 and 0.25 for CT and FDG-PET images, respectively.

### 2.3. Analysis Design

An outline of the design of our analysis is provided in Figure 1. Based on the extracted radiomics features, we used the R package ‘familiar‘ as our modeling framework for predicting LRC [38]. The ‘familiar’ package implements an end-to-end pipeline that preprocesses and clusters features, estimates variable importance, tunes hyperparameters, and builds models based on the exploratory dataset, then evaluates model performance in both the exploratory and independent validation datasets.

We considered all (partial) combinations of modalities (CT, FDG-PET) and timepoints (W0, W2, W3). Feature sets computed from the corresponding images in each subset were concatenated without computing delta features.

For each unimodal experiment, we compared our radiomics models to baseline models that contained the known biomarker tumor volume at each involved timepoint, as this was the only clinical parameter available that could change over time. The non-time-varying clinical parameters age, gender, baseline tumor volume, smoking, alcohol consumption, and different tumor sites (oropharynx, oral cavity, hypopharynx, and larynx) were assessed using univariate Cox proportional hazards regression.

We evaluated two feature selection methods (MRMR [39] and univariate Cox regression) and combined them with the Cox proportional hazards (CPH) model to predict the hazard for a loco-regional recurrence [40].

Models were built in a five-fold cross-validation approach based on the exploratory data. Within each fold, 200 bootstrap samples were drawn from the training fraction of the fold to determine the most important features. Due to the small amount of patient samples available and to avoid overfitting, we limited the maximum number of features each model could contain to four.

For each experiment and both feature selection methods, we further analyzed the feature importance values obtained from the aggregated 1000 bootstrap iterations (200 bootstraps for each of the five folds) and ordered features by the fraction of iterations they appeared among the five features with the best score [41]. We then used the four most commonly occurring features as the final model signature for each experiment. We refit the CPH model on the complete exploration cohort, directly using the obtained features. This model was then evaluated on the independent validation dataset.

In addition, we created and assessed models based on delta features. Delta features were computed by subtracting the feature value corresponding to the earlier timepoint from the feature value obtained at the later timepoint.

### 2.4. Performance Evaluation

Model performance was evaluated in terms of discrimination and stratification ability. Discrimination was measured by the concordance index (C-index), a measure on a scale between zero and one, which evaluates the fraction of patient pairs for which the one with shorter time to recurrence gets assigned a higher predicted risk [42]. A value of one indicates a model with perfect prediction, whereas 0.5 indicates predictions that are essentially random and unrelated to the observed outcomes. In order to measure the ability to stratify patients into low- and high-risk groups with respect to loco-regional recurrence, the median of the model’s predictions based on the exploration data was used as a cutoff point. Each patient was assigned to the low-risk group if the prediction was below the cutoff and to the high-risk group otherwise. A log-rank test was carried out to evaluate differences in the corresponding Kaplan–Meier curves, with *p*-values below 0.05 considered statistically significant.

## 3. Results

We determined and evaluated model signatures for the prediction of LRC for patients with locally advanced HNSCC treated by primary radiochemotherapy. Model signatures were comprised of the four most frequently occurring features identified from the two feature selection techniques (MRMR and univariate regression) in the bootstraps created during cross-validation for each experiment (Appendix A).

Figure 2 and Appendix A contain an overview of the discrimination and stratification performance across all experiments for the MRMR and univariate regression feature selection methods, respectively.

The results when using CT features only are provided in Table 3. The Cox models based on W2, and W2 and W3 combined showed the best discrimination performance on the independent validation data for both feature selection strategies (C-index: 0.78, Table 3). Interestingly, both feature selection methods chose exactly the same features. Patient stratification was statistically significant when using the univariate regression selection method for the combination of W0 and W2 features (p=0.017), although W2 features were exclusively selected as part of the final signature. Furthermore, statistical trends were observed for W2 (p=0.087) and for W0 and W3 combined (p=0.075) for both feature selection techniques.

Combining features from multiple timepoints did not improve upon the model performance of W2. Features from the treatment planning CT did not contain any predictive information when used by themselves (C-index: 0.32 and 0.49 for MRMR and univariate regression, respectively), and reduced model discrimination when combined with features from W2, W3, and from W2 and W3 combined. Models containing tumor volume as the only feature achieved a C-Index of 0.78 on the validation set for W2 and for W0 and W2 combined, but failed to match the performance of the final signature models when combining information from W2 and W3 (C-index: 0.56). Furthermore, none of the available clinical baseline parameters were statistically significantly associated with LRC in univariate Cox proportional hazards regression.

Models based on FDG-PET features obtained from MRMR feature selection showed the best discrimination ability for the combination of W0 and W3 (C-index: 0.70, Table 4). In terms of discrimination, a model that contained tumor volume at W3 as the only feature achieved even better discrimination (C-index: 0.72), but failed to stratify patients into groups of low and high risk of LRC. For both feature selection methods, models derived from W3 features showed increased discrimination compared to W0 models and provided a statistically significant patient stratification in the independent validation (p=0.044 and p<0.001 for MRMR and univariate regression, respectively). However, the obtained risk groups were highly imbalanced in terms of the number of patients, with only one or two patients in the high-risk group for MRMR and univariate regression, respectively (second row of Appendix A).

The results for models that combined CT and FDG-PET features are provided in Table 5. In this scenario, models using features from W3 showed the best discrimination for both feature selection methods (C-index: 0.65 and 0.56 for MRMR and univariate regression, respectively) and obtained statistically significant patient stratification as well (p=0.007 and p=0.044 for MRMR and univariate regression, respectively). Similar to the stratification obtained for FDG-PET experiments, the obtained risk groups were highly imbalanced in terms of the number of patients.

Delta radiomics results are provided in Appendix A. Appendix A contain the corresponding model signatures. For CT-based models, feature changes between W2 and W0 (C-indices: 0.80 for MRMR and 0.69 for univariate regression) as well as between W2 and W0 combined with changes between W3 and W2 achieved the highest C-indices for both feature selection techniques (C-indices: 0.80 for MRMR and 0.71 for univariate regression). Patient stratification was not statistically significant for any of the models. For the FDG-PET based models, feature differences between W3 and W0 showed good discrimination (C-index: 0.73) and significant patient stratification (p=0.002) (with highly imbalanced risk groups) for univariate feature selection, but performed much worse using MRMR feature selection. On the other hand, the model for the combination of CT and FDG-PET based on delta features between W3 and W0 demonstrated better performance when using MRMR feature selection compared to univariate selection (C-index: 0.67 vs. 0.48, p<0.001 vs. p=0.27).

## 4. Discussion

In this study, we explored potential improvements of radiomics-based risk models for LRC prediction of locally advanced HNSCC patients using multimodal (CT and FDG-PET) imaging data obtained at multiple timepoints during treatment compared to only using pre-treatment CT imaging.

In contrast to models trained on pre-treatment imaging features, models trained on features extracted from in-treatment imaging data showed better overall discrimination and patient stratification. CT-based models showed the best discrimination at W2 and at W2 and W3 combined, though patient stratification into groups with low and high risk of recurrence only showed a statistical trend for the former, and was not significant for the latter. Models trained on FDG-PET features created significant patient stratification when imaging data from W3 was used, and showed decent discrimination ability. Combining features from both modalities or from multiple timepoints of a single modality did not result in increased model performance.

Analysis of the model signatures (Appendix A) confirmed the importance of tumor volume as a known biomarker [16,43]. In addition, features such as kurtosis and correlation within the grey level co-occurence matrix (GLCM) for CT were found to contribute to multiple models (Appendix A). Interestingly, while Vallières et al. [9] found correlation in GLCM of CT images to be an important predictor for LRC, tumor volume was not predictive in their analysis.

We did not find evidence suggesting that employing delta features instead of concatenating the features of individual timepoints confers any advantage. While the delta radiomics approach achieved slightly better discrimination using CT feature differences between W2 and W0 based on MRMR feature selection than when combining the features from both weeks (C-index: 0.80 vs. 0.69, Appendix A, Table 3), performance with univariate regression feature selection (C-index: 0.69) was much lower than for MRMR, and was lower than when combining both feature sets (C-index: 0.76, Table 3). Similar observations can be made for the case of FDG-PET features. Here, delta features between W3 and W0 outperformed W3 features with univariate regression feature selection (C-index: 0.73, Appendix A) while providing significant patient stratification (p=0.002). However, using MRMR feature selection with the same dataset produced a model with degraded performance that failed to stratify the patients in the independent valdidation cohort into two subgroups. These findings suggest that the performance of models based on delta features varies more depending on the selected feature selection strategy compared to models working with contatenated features, potentially hinting at less stable results. The qualitative behavior of both methods was nevertheless similar; CT-based models achieved higher discrimination, but had a reduced ability to stratify patients, which was more successfully achieved by models based on FDG-PET features. We observed that models containing FDG-PET features at W3 often produced statistically significant patient stratifications; however, the resulting risk groups were highly imbalanced in terms of the number of patients within each group for the independent validation cohort. We identified inherent statistically significant differences in the values of the used features between the exploration and independent validation cohorts (e.g. Appendix A, last row), leading to deviations in the predictions between cohorts (Appendix A, last row) and subsequently to less balanced strata. Interestingly, for CT feature-based models, stratifications into risk groups was more balanced, though it was rarely statistically significant. In addition, we observed differences in the values of the model features between the exploration and independent validation cohorts for CT features, mostly at W3 (Appendix A, second row), though those did not translate into noticeable prediction differences between cohorts (Appendix A, second row), resulting in more balanced strata. Feature differences may be caused by differences in image acquisition parameters between both cohorts. For CT imaging, the acquisition parameters did not vary much (Appendix A), and we did not need to correct for batch effects. However, it has been shown that the application of harmonization procedures can stabilize and improve machine learning model performance [44,45], and as such we leave this as a direction for future work.

The major limitation of our study is the small number of available patients, which may have caused the issues with lack of statistically significant stratification and highly imbalanced risk groups obtained after stratification described above. The recording of CT and FDG-PET images during therapy is not part of established clinical routine at our institution, which required us to analyze limited data from clinical trials instead. Due to this fact, our exploratory cohort and the independent validation cohort are not completely homogeneous, leading to statistically significant differences in the T stage distribution of patients. The exploratory cohort almost exclusively consisted of patients with T≥3, while that was only the case for about half of the patients in the independent validation cohort. Small sample sizes can make radiomics analyses prone to phenomena such as overfitting. We tried to address this by limiting model signature sizes to a maximum of four features, bootstrapping the feature selection process to ensure that only relevant features enter our models, and choosing a simple and clinically established prediction model (the Cox proportional hazards model). Another consequence of the restricted amount of data is the large confidence intervals for the presented C-indices, limiting rigorous and definitive conclusions. However, we believe our results can be considered a proof-of-concept and contribute further indications to the existing literature on improved radiomics modelling performance when in-treatment imaging data are used.

Leger et al. [31] previously assessed CT imaging only. The dataset used here is a subset of that dataset. Their findings hold true in a qualitative sense, although they show quantitative deviations, which is likely due to the small sample size, e.g., the performance using CT imaging of W2 was slightly higher on our subset than on the data analyzed previously (C-index 0.78 vs. 0.69). Closely related to our patient cohort, Wu et al. [15] analyzed an HNSCC cohort consisting of pre-treatment and mid-treatment CT images of 140 oropharnygeal carcinoma patients for the distant metastasis endpoint, taking GTV and lymph node information into account. Similar to our study, they concatenated features from both timepoints before feeding them to the prediction model (a random survival forest) and observed improved performance compared to a model built from pre-treatment features alone. While they computed selected delta features, they found them to be of little prognostic importance in their analysis.

Only a few publications exist that have investigated longitudinal imaging data for outcome prediction, most of which focused on lung cancer patients. Van Timmeren et al. [33] leveraged longitudinal data obtained for at least four timepoints during treatment for more than 300 patients with non-small cell lung carcinoma (NSCLC) from CBCT images. In their work, the longitudinal evolution of radiomics features was captured by slope coefficients of linear regressions, while we used the feature values directly as inputs to our Cox models. They applied a (LASSO-regularized) Cox model, but did not observe successful validation of their model for LRC that contained pre-treatment as well as longitudinal features. In contrast, we observed improvements when only considering in-treatment features for both modalities. These observed differences could be due to the different tumor entities that were analyzed, or potentially to lower image quality of the CBCTs. Nevertheless, both analyses showed performance drops for CT-based models when including pre-treatment features. Similarly, Fave et al. [46] analyzed LRC for a cohort of 107 NSCLC patients using 4D-CT images. Delta radiomics features capturing the relative change between pre-treatment imaging and end-of-treatment imaging were considered, and a Cox model was applied. While our approach does not rely on delta features, both works suggest that features from imaging obtained during treatment are more prognostic for LRC than those from pre-treatment imaging. A third study on 45 NSCLC patients was recently published, which is similar to our work in terms of sample size [35]. In addition, both works considered CT and FDG-PET imaging. Moreover, the imaging timepoints were pre-treatment and after the third week of treatment, which are comparable to our data. Even though they used a more elaborate multitask ensemble model to predict OS, they found that FDG-PET-based models were able to achieve statistically significant patient stratification, while CT based models failed to. Using a simpler single-task model, they found better performance for W3 models compared to pre-treatment models only for CT imaging, not for FDG-PET, which differs from our results. In line with our findings, they did not observe performance improvements when combining both modalities.

It is known that the fractionated application of radiation treatment affects certain biological properties of a tumor. Examples include changes in epidermal growth factor receptor expression, vessel perfusion, and hypoxic volume [47,48]. In particular, changes in hypoxic volume during therapy have been shown to indicate tumor control [20,22,48]. Similarly, our analysis suggests that imaging obtained during treatment can be assessed to provide improved prediction of LRC. Whether the predictions of our radiomics models are correlated with radiobiological tumor property changes was not investigated; however, this provides an interesting research question for future study.

Despite improved LRC prediction, there are inherent issues around acquiring additional imaging during treatment. First, each additional CT and FDG-PET scan exposes patients to an additional radiation dose, and can be burdensome in other ways. Second, more imaging necessitates additional resources in staffing, consumables, and equipment. These trade-offs must be balanced against the potential benefits of treatment adaptation based on in-treatment imaging.

Improvements in imaging technology might decrease such concerns in the future, e.g., through decreasing costs of FDG-PET or enhanced image quality with low-dose CT scanners such as CBCTs [49,50]. Other innovative technologies, such as MR-guided radiotherapy devices, allow for acquiring MR imaging directly during treatment without requiring additional patient visits for imaging. Even though MR-guided radiotherapy devices are currently not routinely used for treating HNSCC, the predictive potential of machine learning models could benefit greatly from such longitudinally collected series of patient images, opening multiple areas for future scientific work.

In addition, algorithmic developments may help to improve the prediction of clinical endpoints in radiomics studies. Recently, transformer-based deep learning models that have previously led to breakthroughs in natural language processing have been successfully applied to images and sequences of images [51,52,53]. Using an attention mechanism, such models are able to identify prognostic image regions in space (i.e., within a single image) and time (i.e., between imaging timepoints at the same location), which makes them perfect candidates for an application in the context of medical imaging-based survival analysis if sufficiently large longitudinally collected datasets can be obtained.

## 5. Conclusions

In this study, we investigated the prognostic potential of anatomical and functional imaging obtained before and during treatment for locally advanced HNSCC patients treated with primary radiochemotherapy. Compared to pre-treatment imaging, analysis of in-treatment imaging revealed increased predictive performance for both CT and FDG-PET for the endpoint LRC, with CT imaging at week two showing the best overall performance. Combining features from multiple timepoints or multiple modalities had less impact. Further validation of these findings on larger datasets is planned in future work to improve the certainty of the presented results.

## Figures and Tables

**Figure 1 cancers-15-00673-f001:**
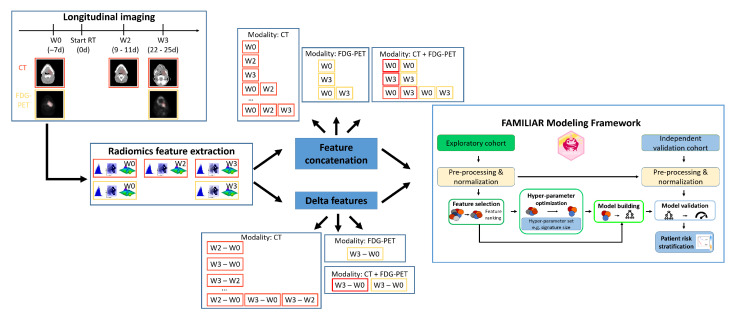
Design of our longitudinal and multimodal analysis. Radiomics features were extracted from a longitudinal collection of CT and FDG-PET scans. Features of different timepoints and modalities were combined and used as input to a machine learning pipeline that fitted Cox proportional hazards models for the prediction of loco-regional control. The 37 patients in the exploratory cohort were used for model building. The fitted models were then evaluated using the data from the 18 patients in the second cohort for independent validation.

**Figure 2 cancers-15-00673-f002:**
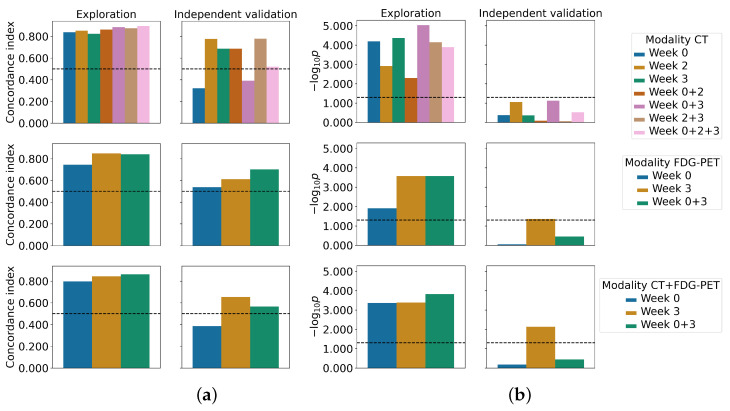
Performance of the final signature models in terms of discrimination (**left**) and stratification (**right**) are shown for both the exploratory cohort and independent validation cohort. The final signature was determined from the MRMR feature importance scores of all bootstraps from the cross-validation. Values above the dashed line indicate non-random predictions and statistically significant stratifications, respectively: (**a**) concordance index; (**b**) log-rank *p*-value, log scaled.

**Table 1 cancers-15-00673-t001:** Median elapsed time of recording of imaging data relative to the start of therapy in days.

Modality	Imaging Timepoint	Exploration	Independent Validation
CT			
	Week 0	−7	−6
	Week 2	11	8.5
	Week 3	25	22
FDG			
	Week 0	−7	−6
	Week 3	25	22

**Table 2 cancers-15-00673-t002:** Patient characteristics of the exploratory and independent validation cohort: *p*-values were obtained using two-sided Mann–Whitney U-tests for continuous variables and $\chi^{2}$ homogeneity tests for categorical variables.

Variable	Exploratory Cohort (n = 37)		Independent Validation Cohort (n = 18)		*p*-Value
	**Median**	**(Range)**	**Median**	**(Range)**	
Age (years)	54	(42–76)	54	(43–67)	0.67
Primary tumor volume at treatment planning (cm${}^{3}$)	31.20	(5.06–141.41)	27.53	(7.02–183.56)	0.087
Follow up time of patients alive (months)	37	(24–70)	62	(8–63)	0.34
Observed loco-regional recurrence time (months)	9	(4–20)	10	(3–23)	1.00
	**Number of Patients**	**(%)**	**Number of Patients**	**(%)**	
Observed loco-regional recurrence	12	(32)	7	(39)	0.86
Gender					
male/female	32 / 5	(86 / 14)	17 / 1	(94/6)	0.67
cT-stage					
T1/T2/T3/T4	0/1/15/21	(0/3/40/57)	0/8/3/7	(0/44/17/39)	<0.001
cN-stage					
N0/N1/N2/N3/unknown	3/6/27/1/0	(8/16/73/3/0)	3/0/13/1/1	(17/0/71/6/6)	0.26
UICC-stage					
I/II/III/IV	0/0/6/31	(0/0/16/84)	0/1/1/16	(0/6/6/88)	0.20
Tumor site					
oropharynx/oral cavity/hypopharynx/larynx	10/12/11/4	(27/32/30/11)	6/6/5/1	(33/33/28/6)	0.91
p16 status					
negative/positive/unknown	29/2/6	(78/6/16)	6/2/10	(33/11/56)	0.37
Pathological grading					
0/1/2/3/unknown	0/0/21/16/0	(0/0/57/43/0)	0/0/6/9/3	(0/0/33/50/17)	0.37
Smoking status					
no/yes/unknown	7/30/0	(19/81/0)	1/7/10	(6/39/55)	1.00
Alcohol consumption					
no/yes/unknown	18/17/2	(49/46/5)	1/7/10	(6/39/55)	0.11

**Table 3 cancers-15-00673-t003:** Final signature model performance (CT): Values in parenthesis denote 95% confidence intervals based on bootstrapping. Values in bold denote the best performance (C-index) and statistically significant stratification (Log-rank *p*-value) for the independent validation cohort. For each experiment, the last row shows the performance of a baseline model that used tumor volume as a feature. The last column provides the fraction of patients assigned to the low-risk group based on the prediction cutoff. The number of patients in the low-risk group of the exploratory cohort is always 18 out of 37, as the median prediction was used as the cutoff.

Experiment	Feature Selection	C-Index		Log-Rank *p*-Value		Patients in Low-Risk Group
		Exploration	Ind. Validation	Exploration	Ind. Validation	Ind. Validation
Week 0	MRMR	0.84 (0.70–0.95 )	0.32 (0.05–0.64)	<0.001	0.42	10/18
	univariate	0.76 (0.62 - 0.89)	0.49 (0.18–0.80)	0.006	0.17	9/18
	– (Volume)	0.62 (0.42–0.78)	0.45 (0.14–0.75)	0.062	0.074	7/18
Week 2	MRMR	0.85 (0.74–0.95)	0.78 (0.54–0.94)	0.001	0.087	10/18
	univariate	0.85 (0.74–0.95)	0.78 (0.54–0.94)	0.001	0.087	10/18
	– (Volume)	0.72 (0.52–0.87)	0.78 (0.49–1.00)	0.020	0.61	9/18
Week 3	MRMR	0.82 (0.69–0.92)	0.69 (0.32–0.95)	<0.001	0.43	11/18
	univariate	0.82 (0.69–0.92)	0.69 (0.32–0.95)	<0.001	0.39	14/18
	– (Volume)	0.74 (0.59–0.86)	0.59 (0.27–0.85)	0.003	0.47	1/18
Week 0 + 2	MRMR	0.86 (0.75–0.96)	0.69 (0.39–0.95)	0.005	0.80	6/18
	univariate	0.86 (0.73–0.96)	0.76 (0.45–0.94)	0.002	0.017	11/18
	– (Volume)	0.71 (0.56–0.86)	0.78 (0.49–1.00)	0.020	0.61	9/18
Week 0 + 3	MRMR	0.89 (0.76–0.97)	0.39 (0.13–0.69)	<0.001	0.075	5/18
	univariate	0.86 (0.71–0.94)	0.40 (0.10–0.66)	0.001	0.075	5/18
	– (Volume)	0.77 (0.61–0.88)	0.62 (0.27–0.88)	<0.001	0.67	1/18
Week 2 + 3	MRMR	0.87 (0.75–0.95)	0.78 (0.48–0.94)	<0.001	0.85	6/18
	univariate	0.87 (0.75–0.95)	0.78 (0.48–0.94)	<0.001	0.85	6/18
	– (Volume)	0.75 (0.59–0.86)	0.56 (0.21–0.81)	0.003	0.47	1/18
Week 0 + 2 + 3	MRMR	0.89 (0.80–0.97)	0.52 (0.20–0.80)	<0.001	0.30	4/18
	univariate	0.86 (0.73–0.96)	0.53 (0.16–0.87)	<0.001	0.075	4/18
	– (Volume)	0.77 (0.62–0.89)	0.66 (0.38–0.88)	<0.001	0.40	2/18

Abbreviations: C-index, concordance index; MRMR, minimum redundancy–maximum relevance; univariate, univariate Cox regression; Ind., Independent.

**Table 4 cancers-15-00673-t004:** Final signature model performance (FDG-PET): Values in parenthesis denote 95% confidence intervals based on bootstrapping. Values in bold denote the best performance (C-index) and statistically significant stratification (Log-rank *p*-value) for the independent validation cohort. For each experiment, the last row shows the performance of a baseline model that used tumor volume as a feature. The last column provides the fraction of patients assigned to the low-risk group based on the prediction cutoff. The number of patients in the low-risk group of the exploratory cohort is always 18 out of 37, as the median prediction was used as the cutoff. * indicates statistically significant *p*-values for the stratification that produced highly imbalanced groups for low and high risk of loco-regional recurrence.

Experiment	Feature Selection	C-Index		Log-Rank *p*-Value		Patients in Low-Risk Group
		Exploration	Ind. Validation	Exploration	Ind. Validation	Ind. Validation
Week 0	MRMR	0.75 (0.59–0.86)	0.54 (0.24–0.85)	0.012	0.90	10/18
	univariate	0.72 (0.53–0.85)	0.51 (0.19–0.81)	0.72	0.78	11/18
	– (Volume)	0.60 (0.41–0.79)	0.49 (0.18–0.78)	0.062	0.17	5/18
Week 3	MRMR	0.85 (0.73–0.94)	0.61 (0.05–0.97)	<0.001	0.044 *	17/18
	univariate	0.88 (0.80–0.95)	0.64 (0.13–1.0)	<0.001	<0.001*	16/18
	– (Volume)	0.74 (0.58–0.87)	0.72 (0.48–0.92)	0.003	0.90	3/18
Week 0 + 3	MRMR	0.84 (0.71–0.93)	0.70 (0.23–1.0)	<0.001	0.36	16/18
	univariate	0.89 (0.79–0.96)	0.42 (0.0–0.87)	<0.001	0.45	12/18
	– (Volume)	0.76 (0.61–0.88)	0.66 (0.37–0.88)	<0.001	0.90	3/18

Abbreviations: C-index, concordance index; MRMR, minimum redundancy–maximum relevance; univariate, univariate Cox regression; Ind., Independent.

**Table 5 cancers-15-00673-t005:** Final signature model performance (CT + FDG-PET): Values in parenthesis denote 95% confidence intervals based on bootstrapping. Values in bold denote the best performance (C-index) and statistically significant stratification (Log-rank *p*-value) for the independent validation cohort. The last column provides the fraction of patients assigned to the low-risk group based on the prediction cutoff. The number of patients in the low-risk group of the exploratory cohort is always 18 out of 37, as the median prediction was used as the cutoff. * indicates statistically significant *p*-values for the stratification that produced highly imbalanced groups for low and high risk of loco-regional recurrence.

Experiment	Feature Selection	C-Index		Log-Rank *p*-Value		Patients in Low-Risk Group
		Exploration	Ind. Validation	Exploration	Ind. Validation	Ind. Validation
Week 0	MRMR	0.80 (0.64–0.94)	0.39 (0.10–0.73)	<0.001	0.66	13/18
	univariate	0.76 (0.60–0.92)	0.56 (0.24–0.85)	0.007	0.89	9/18
Week 3	MRMR	0.84 (0.74–0.93)	0.65 (0.31–0.94)	<0.001	0.007 *	15/18
	univariate	0.84 (0.72–0.93)	0.56 (0.07–0.89)	<0.001	0.044 *	17/18
Week 0 + 3	MRMR	0.86 (0.74–0.94)	0.57 (0.11–0.88)	<0.001	0.36	16/18
	univariate	0.89 (0.82–0.96)	0.38 (0.0–0.84)	<0.001	0.55	11/18

Abbreviations: C-index, concordance index; MRMR, minimum redundancy–maximum relevance; univariate, univariate Cox regression; Ind., Independent.

## Data Availability

The data presented in this study are available on reasonable request from the corresponding author. The data are not publicly available due to local regulations on the protection of patient data.

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
