# Peer review of "Longitudinal and Multimodal Radiomics Models for Head and Neck Cancer Outcome Prediction"

_cancers, 2023, doi:10.3390/cancers15030673_

Round 1
Reviewer 1 Report
In contrast to existing approaches that mostly rely on feature extracted from pre-treatment imaging data, this article focuses on longitudinal and multimodal radiomics, applying machine learning algorithms to mine the association of features from multiple timepoints and multiple modalities with head and neck cancer outcome, with the aim of improving the predictive ability of existing radiomics models. It is helpful and promising to incorporate more available data before the model makes a judgment. Although this article is well organized and has conducted sufficient experiments, the results and conclusions obtained may not be convincing due to data problems. Some comments and suggestions are as follows:
1. In Introduction, the related works could be briefly described to help the reader have a general overview.
2. The samples for exploration and for validation are only 37 and 18 respectively, which is too small. How was the sample split? As can be seen in Table 2, the distribution of samples used for exploration and validation is not balanced.
3. Denial of existing models trained on pre-treatment imaging data requires caution; in my opinion, models trained on pre-treatment imaging data can help in the diagnosis of the cancer, while the addition of subsequent data can help in the prediction of the outcomes.
4. Despite sufficient experiments and rigorous analysis, the conclusion does not convince me. The obtained conclusion was based on these 37+18 samples, with potential problems of overfitting or underfitting. Perhaps different data sets have different conclusions. Besides, the extreme samples may also have a significant impact on the model.
5. Concatenating features at different timepoints may also have a negative impact, and some models that can handle time-series data such as LSTM may be an option.
Reviewer 2 Report
This paper aims to show the predictive ability of a radiomic model for head and neck cancer, with the originality of considering not only baseline but also early follow-up examinations. The analysis protocol is well constructed, including a separate validation cohort, although it may be a little too ambitious for the number of patients included in the study.
The immediate limitation is obviously the number of patients, which results in wide C-index intervals and edge effects on the validation cohorts.
In the end, quantitatively, only the W2 CT model is significantly different from chance on the validation cohort, none of the others reaches significance. In dichotomous models the results are very unbalanced (high risk groups of 1 or 2 patients), so that it is difficult to conclude on the real significance of the results.
1. The significance of the results remains low, especially the added value of radiomic models to clinical models is not studied
2. The GTV contouring method is not sufficiently described. Is the contouring purely manual or with interpolation? Is it done first on the CT alone /a fused PET/CT / a side-by-side ? What software was used?
3. The outcome (LRC) is not enough described. The definition of LRC must be specified (most often it's objective local-regional tumor response in addition to freedom from local progression but it can vary from one paper to another). In particular, the median follow-up duration has to be specified.
4. Technical data are missing and are of upmost importance when considering radiomics
• Was a contrast agent used for CT? It can greatly influence the radiomic parameters. If yes, what is the administration protocol, is it homogeneous across the population ?
• The CT protocol is heterogeneous (z-spacing from 3mm to 5mm can change a lot of thing when resampled voxels ar 1x1x1 mm3), except for week 2 (from which is extracted the only valid model). This heterogeneity might have been detrimental to the other time points. Has a harmonization procedure been considered (Combat algorithm for example) ?
• The PET acquisition protocol is missing (machines used, dose, reconstruction, filters, voxel size). You specify in the discussion that the data are not always complete but maybe you should at least report the data at your disposal.
5. In table 3,4,5 please add the number of patients in both high/low risk group near the log-rank column.
6. Minor comments
• Please correct tables captions so that it can be fully understood without reading the text (explain “–(Volume)” for example or “univariate”). Some results are in bold without explanation.
• Please carefully review the spelling of some words (radiation is spelled for example “radation” or “ratiation” in the text) and use consistently the abbreviations W1, W2… throughout the text
Reviewer 3 Report
The authors combined data from multiple time points and from multiple imaging modalities to improve the predictive power of the radiomic model during the treatment of locally advanced head and neck squamous cell carcinoma (HNSCC). Based on the cross-validation results, 20 feature signatures were obtained, and a Cox proportional risk model was established to independently verify their performance. The authors then assessed discrimination regarding loco-regional control using a C-index and performed a log-rank test to assess risk stratification. The authors concluded that CT was the best discriminating modality, but that none of these models achieved statistically significant stratification of patient risk. While, models based on FDG features produced statistically significant stratification. The radiomics models built by the authors provide a potentially effective tool for personalized treatment strategies. However, there are some concerns that I hope you will be able to answer before the paper is published.
1.HNSCC includes multiple squamous cell carcinomas from the oral cavity, pharynx, larynx, etc., and the tumors at each site have complex typing. The anatomy of their primary region is also complex. The authors included a medium-size cohort, and did not stratify the analysis according to the location, typing and other information, which may affect the effectiveness of the model.
2.MRI may be of equal or greater significance than CT in the diagnosis and prognosis of HNSCC. Therefore, studying the imaging features of MRI at the same time may make the research have stronger application value.
3.Currently, more than 25 studies available in PubMed have explored radiomics' ability to predict HNSCC tumor biology and phenotype, and more than 30 studies have explored radiomics' ability to predict post-treatment events. The classic machine learning approach adopted by the authors in this study does not seem to stand out compared to other studies. Therefore, the innovative or advanced aspects of this research need to be more clearly pointed out.
Round 2
Reviewer 1 Report
My concerns have been addressed. This is a promising work due to its unique focus on the combination of multiple timepoints and multiple modalities, which can motivate the potential improvement of medical models. In addition, even though this work there exists the problem regarding the data issue, the authors have adequately analyzed and explained this in the article.
Reviewer 2 Report
The authors have answered appropriately to the points raised.
Thank you for your work.